# Study protocol for RUFUS—A randomized mixed methods pilot clinical trial investigating the relevance and feasibility of rumination-focused cognitive behavioral therapy in the treatment of patients with emergent psychosis spectrum disorders

Lars Clemmensen[1]*, Christin Nymann Lund[2], Birgitte S Andresen[2], Julie Midtgaard[3,4], Louise Birkedal Glenthøj[1,5]

1 VIRTU Research Group, Copenhagen Research Center on Mental Health (CORE), Copenhagen University Hospital, Copenhagen, Denmark, 2 Mental Health Center Glostrup, Glostrup, Denmark, 3 Center for Applied Research in Mental Health Care (CARMEN), Mental Health Center Glostrup, Glostrup, Denmark, 4 Department of Clinical Medicine, University of Copenhagen, Copenhagen, Denmark, 5 Department of Psychology, University of Copenhagen, Copenhagen, Denmark

* lars.clemmensen@regionh.dk

## Abstract

### Introduction

Psychosis spectrum disorders are characterized by both positive and negative symptoms, but whereas there is good effect of treatment on positive symptoms, there is still a scarcity of effective interventions aimed at reducing negative symptoms. Rumination has been proposed as an important and fundamental factor in the development and maintenance of symptoms across psychiatric diagnoses, and there is a need to develop effective interventions targeting rumination behaviors and negative symptoms in patients with psychotic disorders. The aim of the current study is to investigate the feasibility and acceptability of group rumination-focused cognitive behavioral therapy (RFCBT) in the treatment of young people with psychosis spectrum disorders as well as investigating potential indications of treatment efficacy.

### Methods and analysis

The study is a mixed-method clinical randomized controlled pilot trial with a target sample of 60 patients, who are randomized to either receive 13 weeks of group RFCBT or 13 weeks of treatment as usual (TAU). All patients are examined at the start of the project and at the 13-week follow-up. We will compare changes in outcomes from baseline to posttreatment between group RFCBT and TAU. In addition, qualitative analyzes are carried out to explore feasibility and acceptability and to uncover the patients' experience of receiving the intervention.

**Data Availability Statement:** No datasets were generated or analysed during the current study.

**Funding:** The authors received no specific funding for this work.

**Competing interests:** The authors have declared that no competing interests exist.

# 1. Background

## 1.1 Introduction

Each year between 500 and 1000 people are diagnosed with schizophrenia or other psychotic disorders (collectively termed psychosis spectrum disorders) in Denmark. The disorder most often develops during the late teens and early adulthood [1] and typically debuts in the early 20s. Despite significant progress in treatment methods, including an increased focus on early intervention (counting e.g., the 2-year intensive outpatient treatment, OPUS, which is a permanent Danish psychiatric treatment service for young people with early-onset psychosis) and improved psychopharmacological treatment options, psychosis spectrum disorders are still considered among the most debilitating and expensive disorders in terms of lost workforce and costs of treatment [1–5].

Psychosis spectrum disorders are characterized by positive (delusions, thought disorders, hallucinations) and negative (blunted affect, alogia, anhedonia, asociality, and avolition) symptoms and cognitive disturbances (decreased attention, poor memory, and difficulty in planning). Whereas there is good effect of pharmacological treatment on positive symptoms, there is still a scarcity of effective interventions aimed at reducing negative symptoms and cognitive disturbances [6]. Although there is some evidence of an effect of cognitive remediation and social skills training, there is a need to develop effective interventions for negative symptoms are stressed by the fact that negative symptoms are strongly correlated with vocational and social outcomes, family burden, and quality of life along with being stable over the course of the illness and have a higher impact on later outcome than positive symptoms. Additionally, negative symptoms are strongly associated with depression and anxiety [7]. As much as 75% of patients with psychosis spectrum disorders experience depression and 65% experience anxiety symptoms [8–14]. Depression and anxiety are both strongly associated with relapse of psychosis and impaired social functioning, as well as negatively affected quality of life in people with psychosis spectrum disorders [8–15].

The co-occurrence of multiple psychological symptoms has motivated research aimed at identifying possible transdiagnostic phenomena, i.e., symptoms that are present across disorders and possibly fundamental to the experience of psychiatric symptoms. Rumination, which is characterized by persistent dwelling on negative emotions, problems, and unpleasant experiences, has been proposed as an important and fundamental factor in the development and maintenance of psychiatric symptoms across psychiatric diagnoses, such as depression [16, 17], anxiety disorders [16] and psychosis [18, 19] including both positive [16, 20–22] and negative symptoms [23]. For example, clinical case studies have shown that people with schizophrenia ruminate over the content of their delusions [24, 25], and negative symptoms, including social and emotional withdrawal, which can lead to isolation, and seem to both be worsened by, and increase the risk of, rumination in psychosis spectrum disorders [23].

Rumination can occur due to the individual's attempt and desire to increase self-awareness and solve problems to prevent future mistakes [26]. However, it is suggested that rumination (as opposed to distraction) impairs problem-solving ability and reinforces/supports negative thinking [27] and access to negative memories [28]. The problem remains unsolved and can lead to frustration, stress, and intensify existing problems. Rumination is thus a behavior that induces avoidance behavior that consequently may contribute to increased passivity and worsened anxiety [29]. A self-blaming trait is often seen, which is why it is also a behavior that predicts depressive episodes and their length and severity [30].

Rumination-focused cognitive behavioral therapy (RFCBT) is a type of cognitive behavioral therapy (CBT) targeting rumination. The method differs from standard CBT by not directly challenging the patient's thinking, and by focusing on a change in thinking style (from unconstructive ruminations to more constructive thinking and compassionate problem solving) rather than the actual content of the thinking. Randomized controlled trials (RCT) have shown that RFCBT reduces depressive symptoms and relapse rates in patients with depression [31] and in adolescents and young adults with elevated rumination behaviors. An RCT [32] has shown that a 6-session intervention targeting rumination in patients with psychosis significantly reduced paranoid delusions compared to standard treatment. Apart from this study RFCBT is, however, relatively unexplored in the treatment of psychosis spectrum disorders.

## 1.2 Aim and hypotheses

The aim of the current study is to investigate feasibility, acceptability and potential indications of treatment effect of group RFCBT in the treatment of young people with psychosis spectrum disorders that display symptoms of rumination.

We hypothesize that 1) group RFCBT will be feasible and acceptable for young people with a psychosis spectrum disorder undergoing OPUS treatment 2) RFCBT will decrease negative symptoms, depressive symptoms, rumination behaviour, and increase level of functioning.

We will use the knowledge generated by the present study in the design of a possible subsequent larger fully powered definitive randomized trial.

## 2. Methods and materials

### 2.1 Study design

The study is a mixed methods pilot RCT combining quantitative and qualitative methods. The study will enroll a total of 60 patients from Mental Health Center Glostrup, Mental Health Services in Capital Region of Denmark, which will be randomly assigned to one of the two arms: 1. Experimental group (13 weeks of standard (OPUS) treatment and group RFCBT) or 2. Control group (13 weeks of standard treatment (OPUS)). Patients are assessed at baseline (T1) and at treatment cessation (T2) (see Fig 1: SPIRIT schedule of enrollment). Following completion of post-treatment assessments, the patients in the control group will be offered RFCBT. The study is fulfilling the CONSORT criteria for non-pharmacological treatments and the extension to randomized pilot and feasibility trials.

To best explore the aim and provide the most well-informed basis for possible subsequent protocol for a larger RCT, the current trial combines quantitative and qualitative methods. The value of qualitative research in the development and evaluation of complex interventions is widely recognized in psychiatry research [33, 34], and the combination with quantitative methods will enable not only an indication of the potential clinical relevance of the intervention, but also the organizational and individual prerequisites needed for successful implementation and optimal benefit of the intervention.

The planned study period is from August 1st, 2023—December 31st, 2024.

### 2.2 Participants

The trial is aimed at young people between the ages of 18–35 allocated to OPUS treatment based on a diagnosis of first episode psychotic disorder or schizotypal personality disorder. The following eligibility criteria apply:

| Inclusion criteria |
| --- |
| 1. Age 18–35 |
| 2. Diagnosed with a psychosis spectrum disorder (ICD-10 F2x) |
| 3. At least 8 months left of their OPUS treatment |
| 4. The presence of rumination behavior as assessed by a score of minimum 30 on the Perseverative Thinking Questionnaire (PTQ) [35] |
| 5. Danish-speaking |

| Exclusion criteria |
| --- |
| 1. Substance abuse or positive symptoms that make participation in therapy difficult |
| 2. Severe suicidal thoughts/behavior |
| 3. Not capable of providing informed consent |
| 4. Mild, moderate, or severe intellectual disability (IQ < 70) |
| 5. Planned adjustment of antidepressant and/or antipsychotic treatment (noted in the patient's medical record) |

## 2.3 Sample size

As the overall purpose of the study is to uncover acceptability, feasibility and provide indications of a possible treatment effect. the number of subjects is not based on a calculation of statistical power, but rather on an expectation of how many will be able to include over a project period of 18 months A total of 30 patients receiving the intervention will be compared to 30 receiving standard treatment. A sample size of 30 participants is considered adequate for a pilot-study assessing feasibility, acceptability, and providing indications of efficacy of an intervention [36].

The study is situated at Mental Health Center Glostrup, which has approximately 300 active OPUS patients. In previous trials with group-based CBT in OPUS, it has been possible to recruit a minimum of 10% of the total OPUS sample, thus it is realistic to include the 60 patients in a project period of 18-months.

## 2.4 Randomization

The randomization is done via the computerized randomization function in Research Electronic Data Capture (REDCap) [37] tool in ratio 1:1 on the basis of an uploaded block randomization list (block size 4–6) generated by an external party. In addition, stratification is used to ensure equal distribution of gender between the groups.

## 2.5 Procedure

The study is carried out within the framework of OPUS. OPUS is a 2-year intensive outpatient treatment service for patients aged 18–35 with a psychosis spectrum disorder. Patients meeting the listed eligibility criteria will routinely be approached about participating in this clinical trial. Following inclusion, the participants will have a baseline assessment and be randomized to one of the two treatment arms. Posttreatment assessments will be conducted at treatment cessation (see Fig 2 consort diagram). Among the patients who are allocated to the intervention group, some will also be invited to take part in a qualitative interview. This recruitment strategy presents the risk of healthy volunteer bias. However, our experience from previous similar studies is that this is a minimal limitation in the current setting.

| | STUDY PERIOD | | | |
|---|---|---|---|---|
| | Enrolment | Allocation | Post-allocation | |
| **TIMEPOINT** | $-t_1$ | **0** | Baseline ($T_1$) | Treatment cessation ($T_2$) |
| **ENROLMENT:** | | | | |
| **Eligibility screen** | X | | | |
| **Informed consent** | | | X | X |
| **Allocation** | | X | | |
| **INTERVENTIONS:** | | | | |
| **Group-based rumination-focused cognitive behaviour therapy + opus-treatment (experimental)** | | | ⟶ | |
| **Opus-treatment (control)** | | | ⟶ | |
| **ASSESSMENTS:** | | | | |
| **Demographic characteristics** | X | | X | X |
| **Use of medication** | | | X | X |
| **Addiction** | | | X | X |
| **Self-harming behaviour** | | | X | X |
| **Violent behaviour** | | | X | X |
| **Social Functioning Scale (SFS)** | | | X | X |
| **Perseverative Thinking Questionnaire (PTQ)** | | | X | X |
| **Brief Negative Symptoms Scale (BNSS)** | | | X | X |
| **Ruminative Response Scale (RRS)** | | | X | X |
| **Scale for the assessment of positive symptoms (SAPS)** | | | X | X |
| **Behavior Rating Inventory of Executive functioning (BRIEF)** | | | X | X |
| **Calcary Depression Scale (CDS)** | | | X | X |
| **Group interviews** | | | | X |
| **Semi-structured individual interviews** | | X | | X |

**Fig 1. SPIRIT schedule of enrollment.**

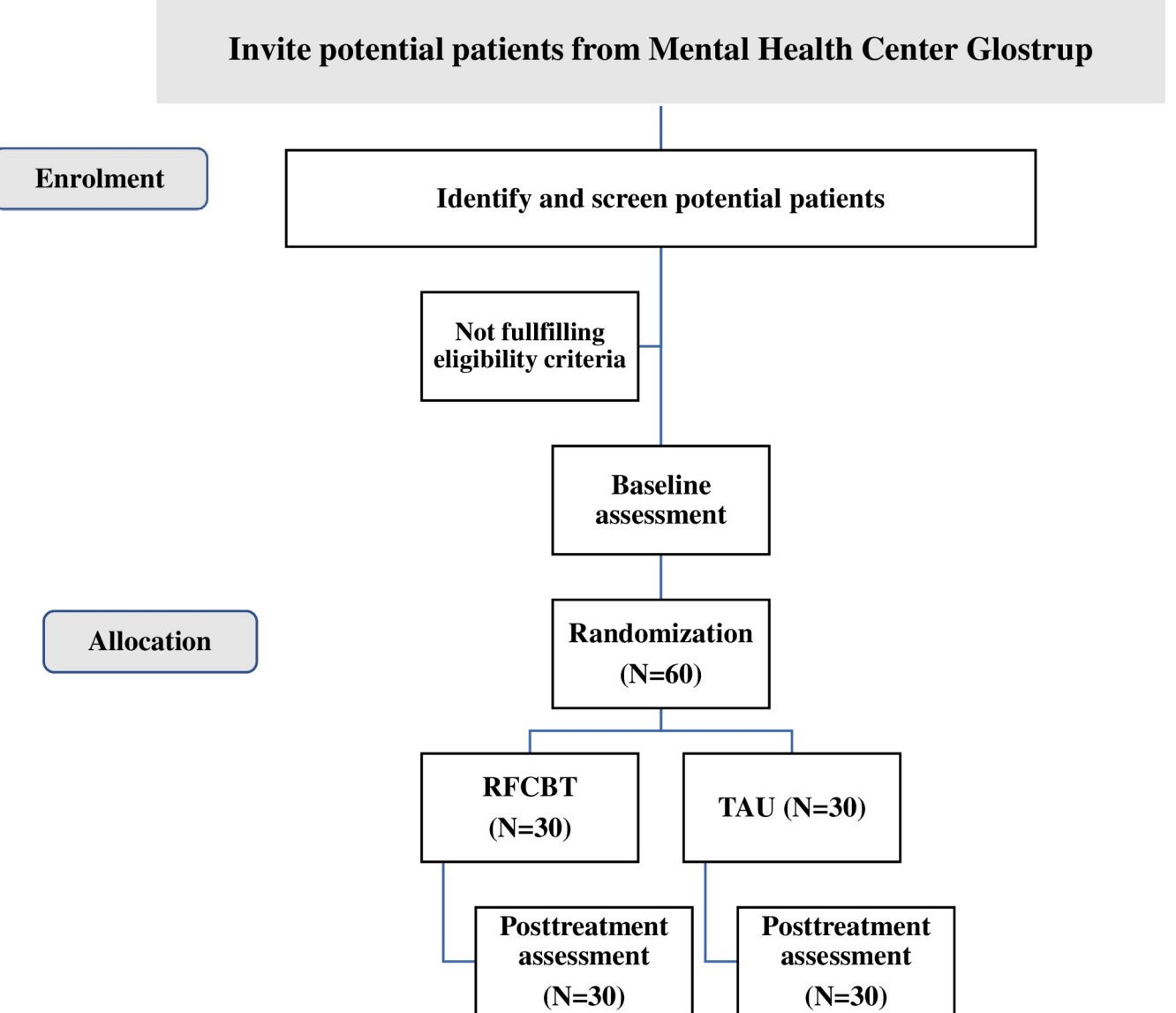

**Fig 2. Consort diagram.** N = number of participants, RFCBT = Rumination-focused cognitive behavioral therapy TAU = Treatment As Usual.

### 2.6 Qualitative sub-study

Individual semi-structured interviews with purposefully sampled participants allocated to the experimental group will be carried out at baseline (focusing on motivation for participation) and repeated posttreatment after approximately three months (focusing on experience of participation). To ensure maximum variation, participants will be sampled according to age, gender and rumination severity, and an interview schedule including open-ended questions and probing is used to support information richness. The information power (i.e., data saturation) is assessed after interviews with 10–12 patients and additional 3–5 interviews will be conducted if needed. For patients who drop out of the intervention before the end of the intervention, the reasons for this are carefully requested. Furthermore, all participants in the intervention group

are invited to a concluding group interview. Consequently, a total of three group interviews will be conducted, with a maximum of ten participants per group."

The purpose of the group interviews is to facilitate discussion and collective evaluation among the participants Finally, interviews with the two therapists responsible for the interventions will be carried out.

## 3. Intervention

### 3.1 Experimental group

The experimental intervention consists of group-based, manualized RFCBT that follows the principal treatment components outlined in the treatment manual developed and published by Watkins [26]. The manual has been translated for use in a Danish context [38]. The treatment comprises 11 groups-session and two individual sessions and is expected to last approximately three months. The treatment is offered as an add on to standard OPUS treatment (2-year intensive outpatient treatment). Before initiating the group sessions, an individual preparatory session (one hour) will take place. This is followed by 11 group sessions of (2 hours/once a week). The intervention ends with an individual session (one hour duration).The therapy includes review and completion of hand-outs, psychoeducation, or dissemination of important messages from the manual, practical exercises, and behavioral experiments (patients cooperate in pairs), guided visualization exercises as well as dialogue and exchange of experiences. RFCBT is based on the theoretical conception, that rumination can be seen as a mental-habit. It uses functional analysis to help people change the habits of rumination and avoidance behavior by identifying triggers and practice alternative behaviors to these cues. Guided visualization is used in the group therapy sessions to motivate and engage the participants in behavior change. The participants conduct small behavioral experiments between the sessions to test the applicability of the alternative behavior in their everyday life. A primary goal is to acquire skills so that one does not automatically fall into the habit of ruminating or avoiding. The intervention takes place in OPUS and is overseen by an experienced psychologist (experiment leader) and a co-therapist.

### 3.2 Control group

Patients in the control group receive standard OPUS treatment and are offered the experimental intervention on completion of follow-up assessment (expected at 3 months). OPUS treatment is handled by an interdisciplinary OPUS team and consists primarily of medical treatment, psychoeducation, training in symptom management and social skills as well as family involvement/ treatment. All patients in OPUS have a contact doctor and a contact person who is responsible for coordinating the treatment and collaborating with municipal bodies.

## 4. Assessments procedures

Questionnaires and measurements are completed at baseline and repeated at treatment cessation after approximately three months. Follow-up clinician-administered measurements are conducted by investigators blind to group allocation. Patients will be instructed not to disclose their allocation prior to post-treatment assessments. In case an assessor is unblinded, the assessment will be conducted by another assessor. The assessors will be psychology master students. All assessors will receiving adequate training prior to conducting the assessments. The assessors will attend regular inter-rater reliability training on key measures in the battery. Quantitative data are collected in REDCap [37] and qualitative data are audiorecorded on a dictaphone and subsequently transcribed ad verbatim and anonymized.

## 5. Outcomes

### 5.1. Feasibility and acceptability

1. Feasibility and acceptability will be defined as:

    a. A recruitment rate of 20% of the eligible OPUS patients.

    b. A therapy completion rate of ≥80% of the recruited participants in the RFCBT group.

    c. 80% of the patients in the RFCBT group reporting a satisfaction rating of ≥20 on the eight item Client Satisfaction Questionnaire [39].

2. Feasibility and acceptability will be further assessed by qualitative interviews with patients in the RFCBT groups, as well as key informants from the staff, including questions on:

    a. Reasons for non-participation and any dropouts during the intervention

    b. The organizational and individual prerequisites needed for successful implementation and optimal benefit of the intervention.

    c. Expenses (hourly consumption, transport, etc.) associated with carrying out the intervention.

### 5.2 Treatment effect

**Primary outcome.**   Negative symptoms:

- Negative symptoms will be assessed using the Brief Negative Symptoms Scale (BNSS) [40]. The BNSS covers six domains of negative symptoms including anhedonia, lack of normal emotional discomfort, asociality, avolition, dampened affect, and alogia. The score is based on 13 questions scored on a 0 (the symptom is absent) to 6 (the symptom is severe) scale. Thus, the total score ranges from 0–78 with higher scores indicating greater impairment/presence of negative symptoms.

**Secondary outcomes.**   Worry/rumination behavior:

- The frequency of rumination is measured using the Perseverative Thinking Questionnaire (PTQ). The PTQ is a self-reported questionnaire and includes 15 questions; each question is scored from 0 (never) to 4 (always) [35], resulting in a total score of 0–60.

- Frequency of rumination in relation to depressive symptoms is measured using the Ruminative Response Scale (RRS). The RRS is a self-reported questionnaire and includes 22 questions that are rated on a 4-point Likert scale ranging from 1 (never) to 4 (always) giving a total score of 22–88 with higher score indicating more severe rumination [41].

    Positive symptoms:

- The presence of positive symptoms is assessed using the Scale for the assessment of positive symptoms (SAPS) [42], that covers the four subdomains hallucinations, delusions, bizarre behavior and linguistic (formal) thought disorder. The scale includes 34 items rated from 0–5, giving a total score of 0–170.

    Function level:

- Level of functioning is measured using the Social Functioning Scale (SFS) questionnaire [43], that contains seven subscales: withdrawal/social engagement, interpersonal

communication, independent-achievement, independent-competence, recreational, proso-cial, and job/employment. The questionnaire has 76 items which are scored from 0 (never) to 3 (often), giving a total score of 0–228, with higher scores indicating better functioning.

  Depression:

- Depressive symptoms are assessed using the Calgary depression scale (CDS) [44] in the form of a structured interview. Items are scored 0–3 on a total of nine questions related to moodi-ness, hopelessness, self-deprecation, self-attributing notions of guilt, pathological guilt, morning depression, early morning awakening, suicide and observed depression. The total score ranges from 0–27 with higher scores indicating higher severity of depressive symp-toms. The scale has been validated for measuring depression in patients with schizophrenia.

  Executive function

- Executive function is measured using the Behavior Rating Inventory of Executive Function (BRIEF) questionnaire [45] consisting of 75 statements that express function in the areas of impulse inhibition, flexibility, emotional control, self-monitoring, initiation, working mem-ory, planning/organization and organization of Materials. Each statement is scored from 1–3, giving a total score of 75–225, with higher scores indicating poorer functioning.

To encourage participants to provide honest and accurate answers and thereby minimize reporting bias, it will be emphasized that their responses will be kept confidential and anony-mous. Self-report questionnaires include a risk of reporting bias. However, we attempt to min-imize this by including indirect questioning and questions about their behavior rather than their beliefs. Indirect questioning refers to a method of asking questions that does not directly inquire about a specific piece of information but rather seeks to obtain that information indi-rectly. That is, instead of asking a straightforward or direct question, individuals use a more subtle or circuitous approach to gather the desired information. This approach is used to create a more comfortable atmosphere, minimizing the chances of biased or inaccurate responses due to the pressure of direct questions. By sidestepping direct queries, it helps foster honesty and reduces reporting bias, where the way questions are framed impacts the truthfulness of answers.

## 6. Analyses

### 6.1 Quantitative analysis

Treatment effect will be analyzed as changes in outcome measures from baseline to post-treat-ment in the RFCBT group using paired samples t-tests or a corresponding non-parametric test such as Pearson's chi-square test. Continuous data are presented as mean ± standard deviation (SD). A two-sided significance level of $p < 0.05$ is used.

  The estimated pooled standard deviation from both groups in this pilot study will be used to inform the sample size calculation for a fully powered RCT.

### 6.2 Qualitative analysis

Data from individual interviews as well as group are analyzed using systematic text condensa-tion or equivalent data-driven thematic analysis [46]. All analyzes are carried out in collabora-tion between several researchers (researcher triangulation) and using NVivo software for qualitative data analysis. A minimum of three members of the research team will be involved in the analysis of the qualitative data. The involved researchers will have diverse educational backgrounds (clinical psychologist, nursing), thereby being able to contribute with different

perspectives in the interpretation of data (i.e., researcher triangulation) to enhance rigor and trustworthiness. Specifically, two researchers will independently code the transcribed interviews and meet with a third researcher to discuss development of higher-order themes.

## 7. Data security, safety, ethics approval, and consent to participate

The study is registered at Danish Data Protection Agency (p-2023-14301) and at clinicaltrial. gov (NCT05851950) and is approved by the Committee on Health Research Ethics in the Capital Region of Denmark (j.no H-23004478). Side effects and adverse events will be monitored and recorded throughout the study period. Any adverse events will be reported to the Committee on Health Research Ethics of the Capital Region Denmark. All patients will be asked to give informed consent after oral and written information about the trial. The patients are informed that they can withdraw their consent at any time, without this having any effect on their treatment or association with OPUS in general. The study is covered by the Danish patient compensation. The study will be carried out on accordance with guidelines from the Danish Committee on Health Research Ethics. In accordance with the Data Protection Act and the Data Protection Regulation, information about the patients will only be collected to the extent that the individual consents to this. Physical media, such as consent forms, will be stored in a locked cabinet in a locked room at Mental Health Glostrup. Electronic data will be stored in REDCap [37].

## 8. Dissemination

Both positive, negative, and inconclusive research results will be published in international peer-reviewed journals. The results will also be presented at national and international meetings and congresses. All information collected in the project and from the journal will be used in the project in anonymized form.

## 9. Patient and public involvement

RFCBT groups have previously been carried out within the framework of OPUS and feedback and suggestions for changes/improvements from patients have been incorporated into the development of the program.

## 10. Discussion

As stated in the background section, development of effective interventions targeting rumination behaviour, negative symptoms, and comorbid depressive symptoms in patients with psychotic disorders is warranted. There is currently only limited knowledge on the effect of rumination-focused therapy in this patient group, but there seems to be great potential for alleviating relevant symptoms and improving the patients' daily life. It is assessed by the research group that it is justifiable to carry out the trial and that the benefits clearly outweigh the few disadvantages that the patients could experience. The results of the trial will be able to give an indication of the effect of rumination-focused therapy, which can subsequently be sought to be confirmed in a randomized controlled clinical trial regarding the possible preparation of well-documented, standardized guidelines for the treatment of this patient group, which can complement the current forms of intervention. several aspects in the current study increase feasibility: Therapists in the study work at the outpatient unit where participants are recruited from and therefore are collaborating closely with the participants' regular therapists. Additionally, therapists in the study are present at the ward on a regular basis and are available to motivate participants to study inclusion and support them during the course. The fact that the

participants already know the outpatient unit is expected to increase study participation and minimize drop-out. The therapists have received comprehensive training in conducting the intervention according to the therapy manual by the first author og this (Morten Hvenegaard) who has also helped in adapting to the specific therapeutic purpose of the study.

## Supporting information

**S1 Checklist.** *PLOS ONE* **clinical studies checklist.** .
(DOCX)

**S2 Checklist.**
(DOCX)

**S1 File.**
(PDF)

**S2 File.**
(DOCX)

**S1 Protocol.**
(DOC)

## Author Contributions

**Conceptualization:** Lars Clemmensen, Christin Nymann Lund, Julie Midtgaard, Louise Birke-dal Glenthøj.

**Methodology:** Julie Midtgaard, Louise Birkedal Glenthøj.

**Writing – original draft:** Lars Clemmensen.

**Writing – review & editing:** Christin Nymann Lund, Birgitte S Andresen, Julie Midtgaard, Louise Birkedal Glenthøj.

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
