## [Decision Letter · Decision Letter 0]

23 Aug 2023

PONE-D-23-16750Study protocol for RUFUS - a randomized mixed methods pilot clinical trial investigating the relevance and feasibility of rumination-focused cognitive behavioral therapy in the treatment of patients with emergent psychosis spectrum disordersPLOS ONE

Dear Dr. Clemmensen,

Thank you for submitting your manuscript to PLOS ONE. After careful consideration, we feel that it has merit but does not fully meet PLOS ONE’s publication criteria as it currently stands. Therefore, we invite you to submit a revised version of the manuscript that addresses the points raised during the review process.

We look forward to receiving your revised manuscript.

Kind regards,

Vincenzo De Luca

Academic Editor

PLOS ONE

Journal Requirements:

5. We note that the original protocol that you have uploaded as a Supporting Information file contains an institutional logo. As this logo is likely copyrighted, we ask that you please remove it from this file and upload an updated version upon resubmission.

Reviewers' comments:

Reviewer's Responses to Questions

**Comments to the Author**

1. Does the manuscript provide a valid rationale for the proposed study, with clearly identified and justified research questions?

Reviewer #1: Yes

Reviewer #2: Yes

2. Is the protocol technically sound and planned in a manner that will lead to a meaningful outcome and allow testing the stated hypotheses?

Reviewer #1: Yes

Reviewer #2: Yes

3. Is the methodology feasible and described in sufficient detail to allow the work to be replicable?

Reviewer #1: Yes

Reviewer #2: Yes

4. Have the authors described where all data underlying the findings will be made available when the study is complete?

Reviewer #1: Yes

Reviewer #2: Yes

5. Is the manuscript presented in an intelligible fashion and written in standard English?

Reviewer #1: Yes

Reviewer #2: Yes

6. Review Comments to the Author

You may also provide optional suggestions and comments to authors that they might find helpful in planning their study.

Reviewer #1: In this study protocol, the authors describe a pilot randomized controlled trial to evaluate the feasibility, acceptability, and potential efficacy of the rumination-focused cognitive behavioral therapy (CBT) on negative symptoms in young adults with psychosis. Rumination-focused CBT has been shown to be helpful for individuals with depression, but less is known about its effect on patients with psychosis, many of whom also experience depression. This trial will contribute new knowledge to the psychiatry field as very little has been studied on whether rumination-focused cognitive behavioral therapy could be a beneficial add-on intervention for patients with psychosis, and how this could improve negative symptoms in these patients.

Major comments:

1. Given this is a single-blinded study (assessor is blinded, but the participants know which group they are in), it is likely that they could have reporting bias when being interviewed for self-reported outcomes via questionnaires. Have you considered any ways to minimize this reporting bias? For example, would it be possible for the control group to have group sessions as well with content different from rumination-focused CBT? Otherwise, there may exist reporting bias when assessing the efficacy of the intervention.

2. How prevalent is rumination in patients with psychosis? If there are patients with psychosis that manifest little rumination, then I suppose they are not the target population of this intervention (as specified in the inclusion criteria). So technically the target population of this trial is patients with emergent psychosis spectrum disorders who have rumination symptoms (a subset), not all patients with psychosis. If so, it'd be helpful to make this clear in this study protocol for the readers.

Minor comments:

1. In the section of “Aim and hypotheses”, quality of life is listed as one outcome to be tested (i.e., RFCT will … increase level of functioning and quality of life). Given there are only questionnaires about functioning, but none for quality of life, I’d suggest removing “quality of life” from the Aim section. Or alternatively (if this is an important outcome to test), you could add a questionnaire on quality of life.

2. Planned study period: will it be 18 months or 12 months? In the section 2.1 study design, it seems that the study will be 12 months (August 1st, 2023 – July 31st, 2024), but it was indicated as 18 months in the section 2.3 sample size.

3. For the line “300 active OPUS patients divided in three teams”: Is there any reason that these patients were divided into different teams and is there any difference between the 3 teams? In the manuscript, it only mentions that gender will be balanced using stratification. Are these different team affiliations also accounted for by stratification, as indicated in the supplementary study protocol?

4. Qualitative interview: can you give some concrete examples on how the sampling will be done to ensure information richness and maximum variation in terms of eligibility criteria? For example, will both the youngest and oldest participants be selected for qualitative interview (e.g., to ensure the variation in terms of patient's age)?

5. Three semi-structured group interviews: will each group interview include 10 people (given there are 30 subjects in total in the intervention group), or will it be group interviews of everyone at 3 different time points?

6. Interviews with key informants: how many staff are planned for the study and how many will be interviewed? If so, is it going to be a random subset?

7. In the 3.1 experimental group section, it says “the treatment comprises 11 groups-session and two individual sessions”, and “the intervention ends with an individual session”. Given there are 2 sessions in total, when is the other individual session, besides the one at the end?

8. Can you briefly explain what are entailed in the behavioral experiments and guided visualization exercise?

9. Research triangulation for qualitative analyses: How will this work? For example, will two researchers work on the same transcribed script and the third researcher could serve as the tiebreaker if there is any discrepancy between the pair of researchers?

10. Limitation in generalizability: it is possible that individuals who consent to participate in the study (either in the intervention group or the control group) could have milder psychosis symptoms. Even though the healthy volunteer bias could be common in RCTs, it is still important to acknowledge this.

Reviewer #2: The current paper presents the protocol of a pilot randomized trial. It targets the feasibility and acceptability of the intervention which is appropriate for such studies. N=30 sample is proposed which is a convenient sample but appropriate for such pilot studies. The paper is very well written, clearly indicating how such feasibility will be determined. I have only two minor comments

1. Please include a CONSORT diagram. While the trial is not complete, a conceptual CONSORT diagram will be helpful for the reader to understand the study.

2. Include a section possibly before the “dissemination” stating the possible or anticipated challenges to feasibility and how the authors will tackle them.

7. PLOS authors have the option to publish the peer review history of their article (what does this mean?). If published, this will include your full peer review and any attached files.

Reviewer #1: No

Reviewer #2: No

---

## [Author Response · Author response to Decision Letter 0]

27 Oct 2023

RUFUS REVIEWER COMMENTS

Reviewer #1: In this study protocol, the authors describe a pilot randomized controlled trial to evaluate the feasibility, acceptability, and potential efficacy of the rumination-focused cognitive behavioral therapy (CBT) on negative symptoms in young adults with psychosis. Rumination-focused CBT has been shown to be helpful for individuals with depression, but less is known about its effect on patients with psychosis, many of whom also experience depression. This trial will contribute new knowledge to the psychiatry field as very little has been studied on whether rumination-focused cognitive behavioral therapy could be a beneficial add-on intervention for patients with psychosis, and how this could improve negative symptoms in these patients.

Major comments:

1. Given this is a single-blinded study (assessor is blinded, but the participants know which group they are in), it is likely that they could have reporting bias when being interviewed for self-reported outcomes via questionnaires. Have you considered any ways to minimize this reporting bias? For example, would it be possible for the control group to have group sessions as well with content different from rumination-focused CBT? Otherwise, there may exist reporting bias when assessing the efficacy of the intervention.

REPLY: We agree that it is important to attempt to minimize reporting bias and we agree that it for that purpose of minimizing reporting bias would be beneficial to have group sessions in both groups. However, group sessions are not part of TAU at the outpatient facility and as we consider it important to be as close to this as possible to have the most relevant comparison, we chose not to include group sessions in TAU. To minimize reporting bias in other ways we have taken other actions and have added the following to the section 5. OUTCOMES: “To encourage participants to provide honest and accurate answers and thereby minimize reporting bias, it will be emphasized that their responses will be kept confidential and anonymous.”

2. How prevalent is rumination in patients with psychosis? If there are patients with psychosis that manifest little rumination, then I suppose they are not the target population of this intervention (as specified in the inclusion criteria). So technically the target population of this trial is patients with emergent psychosis spectrum disorders who have rumination symptoms (a subset), not all patients with psychosis. If so, it'd be helpful to make this clear in this study protocol for the readers.

REPLY: To our knowledge the prevalence has not been systematically assessed but is generally considered high. We agree with the reviewer that the target population could be specified to patients with psychosis presenting with symptoms of rumination, and have added this to the section 1.2 AIM AND HYPOTHESES so it now reads: “The aim of the current study is to investigate feasibility, acceptability and potential indications of treatment effect of group RFCBT in the treatment of young people with psychosis spectrum disorders that display symptoms of rumination”

Minor comments:

1. In the section of “Aim and hypotheses”, quality of life is listed as one outcome to be tested (i.e., RFCT will … increase level of functioning and quality of life). Given there are only questionnaires about functioning, but none for quality of life, I’d suggest removing “quality of life” from the Aim section. Or alternatively (if this is an important outcome to test), you could add a questionnaire on quality of life.

REPLY: We thank the reviewer for pointing out this inconsistency. We have removed “quality of life” from the Aim section.

2. Planned study period: will it be 18 months or 12 months? In the section 2.1 study design, it seems that the study will be 12 months (August 1st, 2023 – July 31st, 2024), but it was indicated as 18 months in the section 2.3 sample size.

REPLY: Thank you for pointing this inconsistency out, we have revised it and it now reads 18 months throughout. We have also changed the study period to August 1st, 2023 – December 31st, 2024

3. For the line “300 active OPUS patients divided in three teams”: Is there any reason that these patients were divided into different teams and is there any difference between the 3 teams? In the manuscript, it only mentions that gender will be balanced using stratification. Are these different team affiliations also accounted for by stratification, as indicated in the supplementary study protocol?

REPLY: The division into three teams is standard procedure at the outpatient facility. We initially considered stratifying by team but decided not to, as no systematic differences can be expected between the teams. On this basis the information on team division is redundant, and potentially confusing, so we have removed it so it now simply reads in the section 2.3 SAMPLE SIZE: “The study is situated at Mental Health Center Glostrup, which has approximately 300 active OPUS patients”

4. Qualitative interview: can you give some concrete examples on how the sampling will be done to ensure information richness and maximum variation in terms of eligibility criteria? For example, will both the youngest and oldest participants be selected for qualitative interview (e.g., to ensure the variation in terms of patient's age)?

REPLY Certainly. We will apply purposeful sampling and aim for maximum variation. We will deliberately select participants who represent a wide range of characteristics to ensure a diverse set of perspectives that can enrich our understanding. We have added the following to the section 2.6 QUALITATIVE SUB-STUDY: “To ensure maximum variation, participants will be sampled according to age, gender and rumination severity, and an interview schedule including open-ended questions and probing is used to support information richness.”

5. Three semi-structured group interviews: will each group interview include 10 people (given there are 30 subjects in total in the intervention group), or will it be group interviews of everyone at 3 different time points? 

REPLY Thank you for turning our attention to this unclarity. We have added the following to the section “2.6 Qualitative sub-study ”Furthermore, we anticipate conducting a total of three semi-structured group interviews, i.e. one interview per therapy/treatment group (3 x n=10). The purpose of the group interviews is to facilitate discussion and collective evaluation among the participants”

6. Interviews with key informants: how many staff are planned for the study and how many will be interviewed? If so, is it going to be a random subset? 

REPLY We have clarified this and added the following to the section 2.6 QUALITATIVE SUB-STUDY: “Finally interviews with the two therapists responsible for the interventions will be carried out.”

7. In the 3.1 experimental group section, it says “the treatment comprises 11 groups-session and two individual sessions”, and “the intervention ends with an individual session”. Given there are 2 sessions in total, when is the other individual session, besides the one at the end?

REPLY: Thank you for pointing out this unclarity. There will also be an initial individual session. We have added the following to the section 3.1 EXPERIMENTAL GROUP: “before initiating the group sessions, an individual preparatory session (one hour) will take place. This is followed by 11 group sessions of (2 hours/once a week). The intervention ends with an individual session (one hour duration).”

8. Can you briefly explain what are entailed in the behavioral experiments and guided visualization exercise?

REPLY: Certainly. We have added the following in 3.1 EXPERIMENTAL GROUP: ”RFCAT is based on the theoretical conception, that rumination can be seen as a mental-habit. It uses functional analysis to help people change the habits of rumination and avoidance behavior by identifying triggers and practice alternative behaviors to these cues. Guided visualization is used in the group therapy sessions to motivate and engage the participants in behavior change. The participants conduct small behavioral experiments between the sessions to test the applicability of the alternative behavior in their everyday life.”

9. Research triangulation for qualitative analyses: How will this work? For example, will two researchers work on the same transcribed script and the third researcher could serve as the tiebreaker if there is any discrepancy between the pair of researchers?

REPLY: we have added the following elaboration to the section on 7.2 QUALITATIVE ANALYSIS: “A minimum of three members of the research team will be involved in the analysis of the qualitative data. The involved researchers will have diverse educational backgrounds (clinical psychologist, nursing), thereby being able to contribute with different perspectives in the interpretation of data (i.e., researcher triangulation) to enhance rigor and trustworthiness. Specifically, two researchers will independently code the transcribed interviews and meet with a third researcher to discuss development of higher-order themes.”

10. Limitation in generalizability: it is possible that individuals who consent to participate in the study (either in the intervention group or the control group) could have milder psychosis symptoms. Even though the healthy volunteer bias could be common in RCTs, it is still important to acknowledge this.

REPLY: We have added the following in section 2.5 PROCEDURE: “this recruitment strategy presents the risk of healthy volunteer bias. However, our experience from previous, similar studies is that this is a minimal risk in the current setting.”

Reviewer #2: The current paper presents the protocol of a pilot randomized trial. It targets the feasibility and acceptability of the intervention which is appropriate for such studies. N=30 sample is proposed which is a convenient sample but appropriate for such pilot studies. The paper is very well written, clearly indicating how such feasibility will be determined. I have only two minor comments

1. Please include a CONSORT diagram. While the trial is not complete, a conceptual CONSORT diagram will be helpful for the reader to understand the study.

REPLY: We agree with the reviewer that a CONSORT diagram will increase readability of the manuscript. In the initial submission we included a flowchart which we have elaborated and named CONSORT diagram in the revision.

2. Include a section possibly before the “dissemination” stating the possible or anticipated challenges to feasibility and how the authors will tackle them.

REPLY: we have added this to the section 11. DISCUSSION: “several aspects in the current study increase feasibility: Therapists in the study work at the outpatient unit where participants are recruited from and therefore are collaborating closely with the participants’ regular therapists. Additionally, therapists in the study are present at the ward on a regular basis and are available to motivate participants to study inclusion and support them during the course. The fact that the participants already know the outpatient unit is expected to increase study participation and minimize drop-out.. The therapists have received comprehensive training in conducting the intervention according to the therapy manual by the first author og this (Morten Hvenegaard) who has also helped in adapting to the specific therapeutic purpose of the study.”

---

## [Decision Letter · Decision Letter 1]

29 Nov 2023

PONE-D-23-16750R1Study protocol for RUFUS - a randomized mixed methods pilot clinical trial investigating the relevance and feasibility of rumination-focused cognitive behavioral therapy in the treatment of patients with emergent psychosis spectrum disordersPLOS ONE

Dear Dr. Clemmensen,

Thank you for submitting your manuscript to PLOS ONE. After careful consideration, we feel that it has merit but does not fully meet PLOS ONE’s publication criteria as it currently stands. Therefore, we invite you to submit a revised version of the manuscript that addresses the points raised during the review process.

We look forward to receiving your revised manuscript.

Kind regards,

Vincenzo De Luca

Academic Editor

PLOS ONE

Journal Requirements:

Reviewers' comments:

Reviewer's Responses to Questions

**Comments to the Author**

1. Does the manuscript provide a valid rationale for the proposed study, with clearly identified and justified research questions?

Reviewer #1: Yes

2. Is the protocol technically sound and planned in a manner that will lead to a meaningful outcome and allow testing the stated hypotheses?

Reviewer #1: Yes

3. Is the methodology feasible and described in sufficient detail to allow the work to be replicable?

Reviewer #1: Yes

4. Have the authors described where all data underlying the findings will be made available when the study is complete?

Reviewer #1: Yes

5. Is the manuscript presented in an intelligible fashion and written in standard English?

Reviewer #1: Yes

6. Review Comments to the Author

You may also provide optional suggestions and comments to authors that they might find helpful in planning their study.

Reviewer #1: The authors have addressed most of the comments well. However, there are a few minor points that I wanted to raise:

1. Multiple titles: On the title page, the title is “study protocol for RUFUS – a randomized mixed method pilot clinical trial investigating the relevance and feasibility of rumination-focused cognitive behavioral therapy in the treatment of patients with emergent psychosis spectrum disorders”. However, the title listed before the Introduction section is “study protocol for RUFUS … investigating feasibility, acceptability and potential efficacy …”. Please select one that more appropriately describes the study.

2. Repetitive phrase in the aim and hypotheses section: it is currently written in the manuscript as “the aim of the current study is to investigate … potential indications of treatment effect ... as well as investigating potential indications of treatment efficacy”.

3. Semi-structured group interviews: It remains unclear whether the authors meant 3 group interviews for the 30 participants in the intervention group (i.e., 10 participants in each of the 3 group interviews). Specifically, the notation of 3*n=10 is confusing since this indicates that the number of participants is not an integer number (i.e., 3.33).

4. Please spell out the whole name for RFCAT.

5. Indirect questioning: it will be helpful to elaborate on what constitutes indirect questioning and how this will minimize the reporting bias.

7. PLOS authors have the option to publish the peer review history of their article (what does this mean?). If published, this will include your full peer review and any attached files.

Reviewer #1: No

---

## [Author Response · Author response to Decision Letter 1]

6 Dec 2023

1. Multiple titles: On the title page, the title is “study protocol for RUFUS – a randomized mixed method pilot clinical trial investigating the relevance and feasibility of rumination-focused cognitive behavioral therapy in the treatment of patients with emergent psychosis spectrum disorders”. However, the title listed before the Introduction section is “study protocol for RUFUS … investigating feasibility, acceptability and potential efficacy …”. Please select one that more appropriately describes the study.

REPLY: Thank you for pointing out this inconsistency. We have changed it so it now reads “Study protocol for RUFUS - a randomized mixed methods pilot clinical trial investigating the relevance and feasibility of rumination-focused cognitive behavioral therapy in the treatment of patients with emergent psychosis spectrum disorders” both places.

2. Repetitive phrase in the aim and hypotheses section: it is currently written in the manuscript as “the aim of the current study is to investigate … potential indications of treatment effect ... as well as investigating potential indications of treatment efficacy”. 

REPLY: Thank you for pointing this out. We have changed it to “The aim of the current study is to investigate feasibility, acceptability and potential indications of treatment effect of group RFCBT in the treatment of young people with psychosis spectrum disorders that display symptoms of rumination.”

3. Semi-structured group interviews: It remains unclear whether the authors meant 3 group interviews for the 30 participants in the intervention group (i.e., 10 participants in each of the 3 group interviews). Specifically, the notation of 3*n=10 is confusing since this indicates that the number of participants is not an integer number (i.e., 3.33).

REPLY: Thank you for bringing our attention to this unclarity. We have changed it in the section 2.6 Qualitative sub-studyso it now reads: “All participants in the intervention group are invited to a concluding group interview. Consequently, a total of three group interviews will be conducted, with a maximum of ten participants per group.”

4. Please spell out the whole name for RFCAT.

REPLY; This is a typo. It is meant to say RFCBT. We have changed RFCAT to RFCBT in section 3.1 Experimental group 

5. Indirect questioning: it will be helpful to elaborate on what constitutes indirect questioning and how this will minimize the reporting bias.

REPLY: We have elaborated and added the following to the outcome section 

"Indirect questioning refers to a method of asking questions that does not directly inquire about a specific piece of information but rather seeks to obtain that information indirectly. That is, instead of asking a straightforward or direct question, individuals use a more subtle or circuitous approach to gather the desired information. This approach is used to create a more comfortable atmosphere, minimizing the chances of biased or inaccurate responses due to the pressure of direct questions. By sidestepping direct queries, it helps foster honesty and reduces reporting bias, where the way questions are framed impacts the truthfulness of answers."

---

## [Decision Letter · Decision Letter 2]

29 Dec 2023

Study protocol for RUFUS - a randomized mixed methods pilot clinical trial investigating the relevance and feasibility of rumination-focused cognitive behavioral therapy in the treatment of patients with emergent psychosis spectrum disorders

PONE-D-23-16750R2

Dear Dr. Clemmensen,

We’re pleased to inform you that your manuscript has been judged scientifically suitable for publication and will be formally accepted for publication once it meets all outstanding technical requirements.

Kind regards,

Vincenzo De Luca

Academic Editor

PLOS ONE

Additional Editor Comments (optional):

Reviewers' comments:

Reviewer's Responses to Questions

**Comments to the Author**

1. Does the manuscript provide a valid rationale for the proposed study, with clearly identified and justified research questions?

Reviewer #1: Yes

2. Is the protocol technically sound and planned in a manner that will lead to a meaningful outcome and allow testing the stated hypotheses?

Reviewer #1: Yes

3. Is the methodology feasible and described in sufficient detail to allow the work to be replicable?

Reviewer #1: Yes

4. Have the authors described where all data underlying the findings will be made available when the study is complete?

Reviewer #1: Yes

5. Is the manuscript presented in an intelligible fashion and written in standard English?

Reviewer #1: Yes

6. Review Comments to the Author

You may also provide optional suggestions and comments to authors that they might find helpful in planning their study.

Reviewer #1: The authors have addressed all the reviewer comments. This study protocol for the pilot randomized controlled trial will contribute to new knowledge on the feasibility, acceptability, and potential effect indication of the group rumination-focused cognitive behavior therapy on negative symptoms in young patients with emergent psychosis spectrum disorders.

7. PLOS authors have the option to publish the peer review history of their article (what does this mean?). If published, this will include your full peer review and any attached files.

Reviewer #1: No

---

## [Editor Report · Acceptance letter]

17 Jan 2024

PONE-D-23-16750R2 

PLOS ONE

Dear Dr. Clemmensen, 

I'm pleased to inform you that your manuscript has been deemed suitable for publication in PLOS ONE. Congratulations! Your manuscript is now being handed over to our production team.

Kind regards, 

on behalf of

Dr. Vincenzo De Luca 

Academic Editor

PLOS ONE